# Development and Validation of an Environmental Health Literacy Assessment Screening Tool for Domestic Well Owners: The Water Environmental Literacy Level Scale (WELLS)

**DOI:** 10.3390/ijerph16050881

**Published:** 2019-03-11

**Authors:** Veronica L. Irvin, Diana Rohlman, Amelia Vaughan, Rebecca Amantia, Claire Berlin, Molly L. Kile

**Affiliations:** College of Public Health and Human Sciences, Oregon State University, Corvallis, OR 97330, USA; Diana.Rohlman@oregonstate.edu (D.R.); Amelia.Vaughan@oregonstate.edu (A.V.); Rebecca.Amantia@oregonstate.edu (R.A.); Berlinc@oregonstate.edu (C.B.); Molly.Kile@oregonstate.edu (M.L.K.)

**Keywords:** water, domestic well, health literacy, environment, scale development, scale diagnostics

## Abstract

In the U.S., privately owned wells are not subject to any regulatory testing requirements. Well owners must have sufficient environmental health literacy (EHL) to understand and interpret information that contain complex terms and labels to manage their water quality. The objective of this paper is to assess the performance and validity of a new EHL screening tool. The Water Environmental Literacy Level Scale (WELLS) is based on the Newest Vital Sign (NVS) and contains six questions on comprehension, calculations and application of information. Content validity was assessed from expert review. Criterion-related and construct validity were evaluated using an online, convenience sample of adults (*n* = 869). Percent of correct responses for items ranged from 53% to 96% for NVS and from 41% to 97% for WELLS. Completion time, mean scores, distributions, and internal consistency were equivalent between both scales. Higher scores suggest higher EHL. The scales were moderately correlated (ρ = 0.47, *p* < 0.001). Kappa agreement was 74%. Bland-Altman plots depicted little mean difference between the scales. Education and income level were positively associated with EHL. WELLS showed criterion-validity with NVS and construct validity with education and income. In practice or research, WELLS could quickly screen individuals for low EHL.

## 1. Introduction

Approximately 34 million Americans rely on privately owned wells to supply their drinking water [1]. The Safe Drinking Water Act (SDWA) applies to all public water systems that have at least 15 service connections or serve at least 25 people per day for 60 days of the year. Unlike people who use public water systems that are regulated by the SDWA, people who rely on private wells must manage their own drinking water supply. Subsequently, a well owner must have knowledge of groundwater hazards, know how to test their well, be able to evaluate the results of water tests, and prioritize resources to adopt appropriate treatment if a hazard is found [2,3,4]. This is problematic because groundwater is frequently contaminated in the United States. Data from the U.S. Geological Survey revealed that 23% of private wells in the United States had one or more pollutant present in excess of its SDWA maximum contaminant level [5]. Well water stewardship surveys report that the majority of well owners do not test their well water even though they are concerned about contaminants [2,3,6,7,8,9]. This is problematic because many harmful contaminants are invisible, have no taste, or smell, and subsequently go undetected by consumers.

People who rely on private wells for their drinking water must be able to seek out independently a variety of information about well maintenance and stewardship. Well owners have to sift through, judge the quality of informational sources, and interpret information about different types of chemical and bacterial pollutants that can pose a health hazard, and devise treatment strategies if a hazard is detected. One barrier that has been identified with regards to improving well stewardship practices, is an individual’s ability to find, or interpret, water quality information [3,9,10]. Water quality information and test results often contain complex terms, labels, and numbers with various confounding units. Thus, environmental health literacy (EHL) levels are likely a contributor to low well water testing and treatment rates. EHL is a relatively new sub-discipline within the field of health literacy [11]. Where health literacy is people’s capacity to access, understand, appraise, and apply information to make decisions concerning healthcare, and disease prevention [12,13], EHL emphasizes an individual’s ability to understand the connection between environmental exposures and human health [11].

Health literacy incorporates several domains. There is literacy, defined as the ability to read and write, and numeracy, the ability to understand probabilistic and mathematical concepts [14]. Numeracy in relationship to health also incorporates the ability to use, interpret and communicate mathematical information and ideas, as well as engage in and manage mathematical demands that arise in different situations during adult life [15]. According to the Department of Education’s National Assessment of Adult Literacy, approximately 35% of the U.S. population has basic or below basic levels of health literacy [15,16]. Limited health literacy is associated with poorer health outcomes and lower adherence to treatment recommendations [12,17,18,19]. Poor health literacy disproportionately affects rural populations and communities of color, with two-thirds of African Americans and three-quarters of Hispanics having limited health literacy [20,21]. Health numeracy skills also tend to be poor in the United States. Data from the National Assessment of Adult Literacy estimates that only 13% of the U.S. population age 16 years and older is proficient in numeracy [14,16]. This assessment data measured how well Americans performed tasks using printed health information and provided a range of tasks to score health literacy on a scale. Based on the tasks, the scale of health literacy could range from below basic or basic based on ability to comprehend information on a pamphlet; intermediate if they could determine appropriate medication use from information on a prescription label; and proficient in health literacy if they could perform calculations of insurance costs from a table of numbers provided. Only 13% of adults were proficient based on performing calculations. Most adults (44–56% depending on age categories) had intermediate levels of numeracy where they could determine appropriate medication dosage from a label. Older aged adults were more likely to score at lower levels of literacy than younger adults [16]. Proper measurement of health literacy and numeracy is important in order to identify individuals who may be at risk for poor health outcomes due to poor health literacy [22].

Experts have advised that health literacy should move beyond the clinic and into public health policy and practice in the community [18,19]. In communities, there are many environmental hazards that are found in private homes, including contaminants that can be found in private well water. Managing environmental hazards is analogous to interpreting medical test results and adhering to clinical recommendations. While health literacy measures have been widely tested [12,13,23], there is a need for EHL screening tools to measure an individual’s ability to comprehend information about chemical risk, and use that information to make decisions that would lead to risk reduction [11]. Subsequently, our goal was to develop and test a screening tool that could be used in research or in practice to screen for environmental health literacy. We named this tool the Water Environmental Literacy Level Scale (WELLS). We developed this scale by adapting information used in domestic well water safety programs regarding arsenic, a common well water contaminant that is a known human carcinogen. We modelled WELLS on the Newest Vital Sign (NVS) assessment tool which assesses health literacy and numeracy [24] Adaptations of NVS have been evaluated and shown to have high internal validity [25]. NVS is a reliable measure of health literacy with good sensitivity and moderate specificity in predicting low literacy and has been found comparable to other health literacy tools such as Test of Functional Health Literacy in Adults (TOFHLA) and Rapid Estimate of Adult Literacy in Medicine (REALM). [26]. The NVS has been used with a range of participants primarily among patient groups representing broad age ranges and diverse ethnic representations. The NVS has begun to been assessed in non-medical settings such as churches, libraries, community centers, participant homes [27].

Our objective was to assess the performance of WELLS using scale diagnostics and with content, criterion-related, and construct validity. Specifically, we compared WELLS to NVS on: (1) absolute values, (2) ranking ability in tertiles, and (3) Bland-Altman plots. Construct validity compared both WELLS and NVS with socio-demographics (i.e., education, income) known to relate to health literacy and with different experiences with home hazard stewardship (i.e., prior experience with septic tank or domestic well) which theoretically is expected to positively associate with higher environmental health literacy.

## 2. Materials and Methods

The WELLS scale was developed using guidelines from DeVellis [28]. Scale development steps included: (1) determine the construct to be measured; (2) generate an item pool; (3) determine format for measurement; (4) have initial items reviewed by experts; (5) consider validation items; (6) administer items to a developmental sample; (7) evaluate items; and (8) optimize scale and length. Content validity was assessed through scale adaptation from the NVS and from expert review. Criterion-related and construct validity was measured using an online, convenience sample of adults in the United States.

### 2.1. Scale Development and Item Generation

We determined that our construct to be measured was environmental health literacy, with a focus on numeracy and decision-making. We wanted a scale similar to the NVS in terms of constructs covered, number of items included, and length of time to complete. We used an excerpt from health education materials for well stewardship that was similar to the ice cream label used in the NVS to include in WELLS. We selected a figure that listed safe water use for a range of test results following an arsenic test (See Figure 1). This figure is comparable to the ice cream label, where participants have to comprehend information provided in a table in order to perform calculations and make decisions. The arsenic figure has been used in health education material provided by the Oregon Health Authority [29].

We adapted questions and scoring to mirror the NVS. Specifically, we asked a total of 6 items that included comprehension, calculations, and application of information for a hypothetical decision. One point is awarded for each correct response. The scale is summed with a range from 0–6 with higher scores corresponding to higher levels of environmental health literacy.

### 2.2. Expert Review and Inclusion of Validation Items

The table, questions, and response options were reviewed by experts in environmental health at Oregon State University, College of Public Health and Human Sciences and from the state’s public health authority—Oregon Health Authority, Public Health Division, Environmental Public Health Surveillance Program. Individuals from these groups either work with programs to improve environmental health at the individual and community level, or to improve public health messaging. Experts commented on the relevance of items to each construct, clarity of wording and response options, acceptability for use in the field, and improvements to capture meaning of scale. Our experts were also provided with a copy of the NVS scale and we requested their input on variables to include in our survey to test for construct validity.

### 2.3. Survey Administration to Determine Criterion-Related and Construct Validity

This study used a convenience, non-probability sample to test the validity of WELLS. A survey was constructed that included both health literacy scales (NVS and WELLS) and variables needed for construct validity. Inclusion criteria was adult age 18 years and over, living in the United States, and able to read English. Our survey was administered online to adults on Amazon Mechanical Turk (MTurk). MTurk is an online Web-based platform for recruiting in which participants are paid a small amount of money for completing tasks posted online. MTurk is an online Web-based platform for recruiting and paying subjects to perform tasks. Relative to other experimental pools. Participants or workers on the site (aka “Turkers”) log into the MTurk website are choose opened job tasks listed on a job posting site [30]. On MTurk, participants (“workers” aka “Turkers”) review postings for human intelligence tasks (“HITs”) by title, keyword, reward amount, availability. They are paid upon successful completion of the tasks at a piecework rate [31]. One could compare MTurk to a job wanted board where different organizations list their job openings, duties and pay rates. Except for MTurk, the job board is an online platform and the jobs are simple and quick and involve human coding or interpretation (i.e., surveys, coding labels, etc.). MTurk participants are unsupervised and anonymous, complete surveys or taks in unknown locations, and are motivated by financial incentives [32]. A study by Chandler at al (2014) found that most workers completed the task at home and alone; they were often engaged in other activities simultaneously such as watching TV, listening to music or using social media [31].

Amazon does not provide any information about the numbers or demographics of the MTurk workers population. However, studies conducted by outside researchers of MTurk participants have shown that the number of Turkers remains stable overtime and consists of 100–200,000 or more workers with an average half-life of 400 days [33]. The majority of MTurk workers come from the U.S. (75%) and then India. MTurk samples have been shown to be gender-balanced, geographically distributed, racially and ethnically representative, and having a mean age in the mid-thirties [34,35]. Among U.S. workers, the sample is slightly more female than male [33], more white [31], notably younger and more ideologically liberal than the general U.S. public [36]. MTurk subjects are often more representative of the general population and substantially less expensive to recruit [36].

MTurk samples have produced valid and reliable data on a variety of health literacy measures [37,38] and allow for the recruitment of relatively large samples quickly and for minimal cost as compared with other online recruitment strategies or in-person convenience sampling [36,39]. MTurk participants have shown to be more attentive to instructions and survey questions than other convenience samples [32,36] and provided higher quality data in terms of fewer missing responses and more thoughtful as compared with other online counterparts (Google, Facebook) [39]. Concerns about repeat survey taking is minimal and has not been shown a large problem in the MTurk subject pool [36]. Recruitment on MTurk was two-stage. Participants first completed a pre-survey with filter questions that included age, gender, zip code, and an attention filter question to ensure responses were not due to automated robot response. Upon completion, participants were offered compensation of $0.03 Amazon credit following MTurk compensation guidelines. Respondents who answered the filter questions correctly were invited to the full-survey via email with a unique URL. The full survey includes questions on environmental health literacy, Newest Vital Sign, word recognition items (not presented here), information seeking behavior (not presented here), and demographics. The full survey took approximately 20 minutes to complete and participants were compensated with $1.00 in Amazon credit. Prior to online testing, the potential survey items for WELLS were reviewed by a sample of well owners (*n* = 5). In addition, staff from local, regional and state domestic well programs reviewed the scale prior to implementation. The survey was pilot tested with individuals online to test for timing, non-response or unsure response to items, and appropriateness of response options. A total of 166 participants piloted the survey between 24–27 October 2016. Distributions of each item and response option were reviewed for lack of response or abundance of “don’t know” selected. The timing and overall response rate was sufficient. Minor changes were made to response options, but not scale items. Data collected from this pilot test were used to inform the scale development, but not scale validation, and are not included in the Results section of this manuscript.

The revised survey was administered to the final sample to validate. Data were collected between 21–25 November 2016. A total of 1168 initiated the filter survey and 1153 completed the filter and were invited to the full survey. The filter survey took on average 7.3 seconds (9.9 seconds standard deviation) to complete. Of the 1153 who completed the filter survey and were offered the full survey, 911 started and 869 completed the survey (See Table 1).

### 2.4. Measures

#### 2.4.1. Water Environmental Literacy Level Scale (WELLS)

WELLS is a six-item scale and each correct response was awarded one point. Higher scores relate to higher environmental health literacy. Figure 1 is the image that participants referred to when answering questions. Table 2 contains the exact wording of the questions and the number correct. Appendix A reports the specific response options as well as their distribution.

#### 2.4.2. Newest Vital Signs

NVS asks participants to answer questions based on an ice cream nutrition label [24]. Scores are assigned from zero to six, with one point for each correct answer. Higher scores relate to higher health literacy. Newest Vital Signs has been shown to be an effective quick screening tool, it is available in English and Spanish, and it can be used to evaluate health literacy for populations with low literacy levels [24,25]. Table 2 contains the exact wording of the questions and the number correct. Appendix A reports the specific response options as well as their distribution.

#### 2.4.3. Well and Septic Tank Use

Survey respondents were asked if they had ever lived, or currently live, in a home with a private well or a septic tank. Response options were all binary yes/no. These responses were used to test the construct validity of WELLS measures. Specific wording was, “Have you ever lived on a private property with a well? If yes, is there a private well on the property where you currently live?” and “Have you ever lived on a property with a septic tank? If yes, is there a septic tank on the property where you currently live?”

#### 2.4.4. Sociodemographics

Age, gender, race/ethnicity, education, income, own or rent home, number of years in residence, and number of people in household were included using standardized wording from the Centers for Disease Control and Prevention’s Behavioral Risk Factor Surveillance System [40].

### 2.5. Statistical Analysis

Univariate analyses and histograms were used to evaluate the distribution and variance in all items. Scale diagnostics and psychometric properties were examined for both NVS and WELLS scales by performing a factor analysis, reviewing a scree plot, and calculating the internal consistency. Factor analysis incorporated an orthogonal varimax rotation. Factors were retained with Eigenvalues greater than 1.

Next, statistical tests assessed criterion-related validity between the WELLS and NVS. Kappa statistics and correlations compared the relationship between the absolute values. Ranking ability was tested by reporting the percent agreement of the two measures that fall within the same tertile. Bland and Altman limits of agreements (LOAs) between the two measures were calculated because a correlation is not a sufficient statistical test when comparing a new measurement to an established one [41]. The correlation measures strength of the relation between the two measures, but not the agreement between them. As such, reliability and validity studies should construct the limits of agreements (LOAs) between the two measures. LOAs provide a numeric distance for how far apart the new method is from the established method. The LOA provides information on the magnitude of error between both measures, the direction of bias, and whether or not bias is constant across levels of measures. The LOA is calculated by first determining the differences in the absolute values of the measurements by the two different methods (NVS-WELLS) for each individual and then calculating the mean and standard deviation of the group. The 95% confidence interval of the LOA represents a range of values within which 95% of all differences are expected to fall. Using the SD of the differences between the methods, 95% LOA are calculated as +/−1.96 from the SD of the differences. Computations followed Bland and Altman recommendations [42]. Bland-Altman Plot is provided which plots each individuals’ data with the average of the two scores on the x-axis and the difference between the two scores on the y-axis. This plot depicts at which scores the two scales are similar or diverge. Lastly, we tested construct validity comparing absolute scores on the environmental health literacy scales between different household characteristics (i.e., own home, own well, children in home) and socio-demographics (i.e., education) theoretically related to environmental health literacy. Bi-variate analyses included *t*-tests, chi-square and ANOVA depending on the format of the variables. All analyses were conducted using StataSE version 13 (StataCorp, College Station, TX, USA).

### 2.6. Research Ethics

All participants agreed online to the informed consent prior to completing the survery. A separate consent form was included on the first page of the webpage prior to answering any questions. However, participants did not sign a consent form. Rather, they agreed to participate by clicking the “Next button” on the webpage.

There was a separate consent for the screening questions. Participants read the explanation of research document on the first page of the survey and clicked “Next” to indicate that they agree to participate prior to the beginning of the study. Participants who completed and passed the screener were invited to participate in the full survey. Eligible participants received an email with a link to the full survey. The consent form for the full study appeared on the first page of the webpage. Participants read the explanation of research document on the first page of the survey and clicked “Next” to indicate that they agree to participate prior to the beginning of the study.

All data collection procedures were approved by the Institutional Review Board at Oregon State University (IRB protocol# 7299).

## 3. Results

Of the 911 people who initiated the survey, 95% (*n* = 869) completed the full survey that was used to test the scale performance. Participant characteristics are described in Table 1. In the Appendix, Figure A1 displays the location of participants who completed the survey. The majority of people who filled out the survey were female (57.5%). Most participants were white (77%), followed by Latino (6.9%), Black or African American (6%), and Asian or Asian Indian (5%). Almost half of the participants were college graduates (51.8%), with 34.4% having completed at least some college. Almost one-third of participants (35%) identified as low-income, earning less than $35,000, and only 24% earned over $76,000 a year. The amount of homeowners versus renters was relatively even, with 49.3% owning their homes, 43.8% renting, while 6.8% had some other kind of living arrangement. The most common household size was two people (30.5%), followed by three people (20.5%), and one person (18.5%). Only 4.9% had more than six people in their household. Approximately one-third of the participants (37.2%) reported having ever lived on a property with well and 11.4% currently living on a property with a well. More participants reported living on a property with a septic system (ever using a septic system, 51.1%; and currently using a septic system 23.2%).

### 3.1. Univariate and Scale Diagnostics

Survey respondents completed each scale in an average of 2 min. Table 2 reports the specific item wording with percent correct for each item. Percent of correct responses for the individual items ranged from 53% to 96% for the NVS scale and ranged from 41% to 97% for the WELLS scale. When correct items were summed within scales, their mean scores and distributions were equivalent with both scales reporting a mean score of 5.0 and standard deviations for 1.2 for NVS and 1.0 for WELLS scales. Histograms are available in the Appendix, Figure A2 for each scale, and depict a right-skewed distribution. A higher proportion of respondents had higher scores than lower scores. Internal consistency was similar between the two scales. Cronbach’s coefficient alpha was 0.62 for NVS and 0.51 for WELLS. The factor analysis for each scale revealed only 1 factor with an Eigenvalue greater than 1, and thus, suggested retaining one factor per scale. We have included a scree plots for each of the scales in order to depict the drop in Eigenvalue magnitude (or the “elbow” of the scree plot) after the first factor but before the second factor. Plots are available in the Appendix, Figure A3.

### 3.2. Criterion-Related Validity

The two health literacy scales reported a moderate, but significant correlation with each other (ρ = 0.47, *p* < 0.001). Kappa agreement was 74% and the kappa statistic was significant at 0.29 (standard error of 0.03). Continuous scores were separated into tertiles: low (score of 0–2), medium (score of 3 or 4), and high (score of 4 or 6) for each scale. For both scales, most participants scored in the top tertile (77% for NVS, 77% for WELLS), followed by the medium tertile (17% for NVS, 19% for WELLS) and then the lowest tertile (6% NVS, 4% WELLS). Six hundred and forty-three participants were classified within the same tertile for both scales. Therefore, our scales had a tertile agreement of 74.0% (643/869). Figure 2 displays the Bland-Altman plot for NVS and WELLS scales. The mean difference between the scores of the two scales was 0.012 (SD = 1.16) and the 95% lower (−1.15) and upper (1.18) limits of agreement.

### 3.3. Construct Validity

Construct validity was determined using bivariate analysis between the health literacy scores and theoretically-related variables (Table 2). As expected, education and income level were positively related and statistically significant with both WELLS and NVS scales. The highest income bracket in the survey (over $76,000) was related with the highest WELLS and NVS scores. The respondents who had some high school were found to have the highest scores, but there were also only two respondents in that category. The rest of the education categories showed higher WELLS and NVS scores for higher education levels. There was no relationship to environmental health literacy and having experience with either a well or septic system, even though our screening tool was based on information developed for people who use a private well.

### 3.4. Sensitivity Analysis

Given possible different interpretations to WELLS item b, we conducted two sensitivity analyses. First, we omitted item b from the total score. Second, we re-coded item b to reflect a more inclusive interpretation of what constituted safe domestic use. The results of these sensitivity analysis are presented in Table 3. Omitting item b slightly improved the reliability of WELLS compared to NVS. Yet, both re-analyses found similar relationships and statistical significance with sociodemographic factors where people who had a lower income, lower educational attainment or self-identified as Latino had lower WELLS.

## 4. Discussion

We developed and assessed the content, criterion-related, and construct validity of Water Environmental Literacy Level Scale (WELLS), a new environmental health literacy screening tool. This measure was adapted for well water stewardship from the Newest Vital Sign (NVS), a gold standard health literacy assessment tool. WELLS performed similarly to NVS. However, criterion-related validity does not imply a causal relationship, but rather documents the strength of the relationship between the two measures. We would expect that the two measures (WELL and NVS) would correlate because of shared variance from similar methodologies and shared concepts of literacy and risk communication. However, it was possible that differences would be observed between the two scales because individuals could be more familiar with comprehending nutrition labels than the risk posed by arsenic exposures. However, very little mean difference was observed between the two scales at the lowest (1) and highest (6) ends of the scale scores. Most of the variance between the two scales occurred when participants scored in the middle of the range.

While we found the WELLS scale to show strong criterion-related validity with NVS, we did not consistently observe the expected construct validity. Construct validity is the theoretical relationship between the score of the scale with other established measures. In this study, we chose demographics of education and income which have been shown to be related to health literacy [18,19]. We added variables of prior well or septic system use as markers of experience that should theoretically relate with environmental health literacy. As expected, we saw a strong positive relationship between education level and income with higher WELLS scores. We did observe two individual cases without a high school degree who scored high on both health literacy scales, but aside from those two cases the pattern was as expected. Yet, we did not see any relationship between prior or current experience with a well or septic system. This unexpected finding could indicate that even though an individual had lived with a well or septic, they might not have engaged in any of the risk mitigation activities because they were too young, were not head of the household, or recently purchased the home and had the environmental review performed by professionals during the real estate transaction. In addition, only a small percentage of well owners report that they routinely test and/or treat their wells [2,3,4,6,43,44,45]. Therefore, our expectation that experience with well ownership or septic tanks would improve environmental health literacy may be unfounded.

We have limited variance in our scales. Over 2/3 of the sample were in the higher levels for both WELLS and NVS. In addition, for both WELLS and NVS, only two items had variance in number of incorrect responses. For 3 items, the majority of the sample (over 95%), answered them correctly. This finding matches other validation or correlational analyses seen in other studies using NVS. In samples with higher education, the scores on NVS tend to be skewed higher (stronger literacy). In several studies with English-speaking samples, 40–80% of the samples scored at the top two levels of literacy using NVS scales (a score of a 5 or 6) [25,27,46,47]. Most of these previous studies did not list the correct response rate for each individual item. One study did report the correct responses by individual items. Similar to our study, their study found only two items that showed low correct responses and they were questions that required more calculations or interpretations of calculations [48]. Even among samples with low average literacy scores across the population, approximately 35% had a perfect score on the NVS (6 out of 6) and 50% scored a 5 out 6 [49].

One of the major limitations of this study is that the survey participants were recruited through MTurk, and we assume that this population will be younger and more educated than the general public [33]. In our inclusion criteria, we did limit the survey to MTurk workers who had a track record of successful completion of tasks on MTurk. Although the identity of workers is typically not known, their responses across different HITs can be monitored, integrated, and managed. These “Super Turkers” have greater experience and more savvy with psychological experiments which may lessen the generalization of our sample [30].

Therefore, we also administered the survey in paper format at local community events. Because of the small number of surveys collected, many cell sizes were zero in the data collected from paper administration which inhibited us from comparing to the MTurk data. The first event was the annual small farms conference in Oregon which provides education sessions in English and Spanish geared towards farmers, agricultural professionals, food policy advocates, students, and managers of farmers markets. The second event was the Festival Latinos in the Willamette Valley, Oregon which provides free bilingual health and educational resources as well as health screenings. At both events, an outreach table was set-up and participants who walked by the table were recruited who met the same eligibility requirements as the MTurk survey. Only English-language surveys were available at the Small Farms Conference while surveys were available in both English and Spanish at the Festival Latinos. The paper and pencil version took approximately 4–7 minutes to complete. A total of 13 surveys were completed at the Small Farms Conference with a mean of 4.0 for both surveys (NVS SD = 1.11 and WELLS SD = −0.93). Respondents from the Small Farms were educated (92% had at least some college) and had more experience with wells (46% currently, 62% ever) and septic systems (54% currently, 69% ever), yet scored poorer on the WELLS and NVS scales. At the Festival Latinos, a total of 10 surveys were completed, with a mean of 3.8 for NVS scale (SD = 1.9) and mean of 4.1 on WELLS (WELLS SD = 1.8). We did not have a large enough sample size from the smaller conferences to compare to our larger survey. However, the scales were acceptable when tested with different populations in a different mode of delivery.

Although we did test the survey with owners of small farms, we did not specifically pre-test the instrument with only well owners and we did not perform any cognitive testing. We relied on our panel review who worked directly with well owners. In addition, it was difficult to collect a large number of only well owners to test the survey. On MTurk, we could have limited the survey to only well owners, but there would have been no way to validate this inclusion criteria. Finally, the general public may not be as familiar with the units used to characterize the concentration of arsenic in water (e.g., ppb and mg/L). Including definitions and explanations for these units could further reduce variation between NSV and WELLS.

Finally, it is possible that respondents may interpret “safe for domestic use” (e.g., WELLS item b) differently. While our sensitivity analysis found that response to this particular item did not change the overall sensitivity of the WELLS scale, the phrasing of this item may be problematic.

Environmental health literacy is a growing area of interest and screening tools are needed to direct resources to populations who are most at need of water mitigation services. Currently, federal, state, and county public health agencies promote non-regulatory water safety programs that distribute risk information, provide hotlines for questions, and (sometimes) subsidize the cost of water testing [50,51,52]. While these programs are capable of increasing awareness of the homeowners responsibility to test their drinking water and increasing knowledge of local hazards [6,53], the majority of well owners do not test their water even though they are concerned about environmental pollution, nor do not treat their water even if a contaminant is present [3,10,50,52,54]. For example, 27% of well owners in Maine did not act after arsenic was detected in their well [23]. The primary reason given for not testing or treating drinking water is that it smelled, looked, and/or tasted fine. [2,3,6,7,8,9]. This is problematic. Arsenic, nitrate, and lead are odorless, tasteless, and invisible. Additionally, there is a long latency period between the cumulative exposure and adverse health effects making it more difficult for people to attribute their health outcomes to an environmental source [55]. Finally, free-testing programs reported that discounted or free water testing was more likely to be used by people with higher (not lower) socio-economic status [2,52,56,57] which increased the health inequities not ameliorated them. Programs aimed at clean drinking water should focus on both health literacy needed to test their well and health literacy needed to treat their well. A screener for environmental health literacy is needed that can easily discriminate low versus high literacy using a tool that is easy to administer, quick to complete and score. Our initial attempt at this screener revealed that we could adapt NVS for environmental health literacy. However, our scale should be revisited and tested in different samples.

Future studies should rephrase the question wording and prompt. We chose the sample material for the WELLS measure from standard information that is available in our state to well owners. However, the clarity of our material itself and our specific questions should be revisited. Specifically, the use of parts per billion (ppb) may be an inappropriate choice of units because many of Americans are unfamiliar with this unit.

On the other hand, we constructed the assessment items to not need to be familiar with ppb. Secondly, the word “domestic” might be confusing for individuals and connote different meanings to different individuals based on if they have experience with farming or other agricultural work. We suggest changing the word “domestic” to “household” use in both the material and question prompt. An alternative suggestion would be to rephrase item 2 to ““How many ppb of arsenic makes water unsafe for any domestic use?” In addition, we plan to test these revised samples with specific well owners using cognitive testing and formative research and conduct a study with more diverse samples.

## 5. Conclusions

The WELLS measure could discriminate participants with low environmental literacy from those with higher environmental literacy. This screening tool is easy and quick to complete and similar completion time (<3 min) as seen in systematic reviews of NVS [27]. It was well received by participants and could be administered electronically as well as in person. Health promotion is often the only course of action for reducing an individual’s exposure to unregulated toxins or pollutants that occur in the home environment. This screening tool does not need to be limited to individuals with a well. Currently, most water utilities are not able to meet the SDWA. EHL would be important to screen for and tailor materials or programs for individuals with a range of source of drinking water including wells and water utilities. Screeners such as WELLS could be used by public health practitioners to identify individuals who might require additional services and support for public health programming that is trying to address environmental health disparities.

## Figures and Tables

**Figure 1 ijerph-16-00881-f001:**
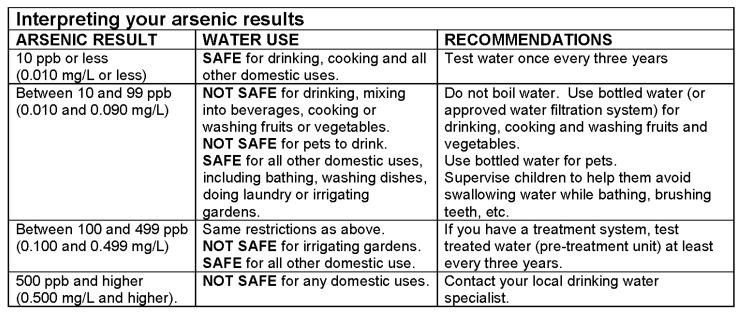
Visual aid used in Water Environmental Literacy Level Scale (WELLS).

**Figure 2 ijerph-16-00881-f002:**
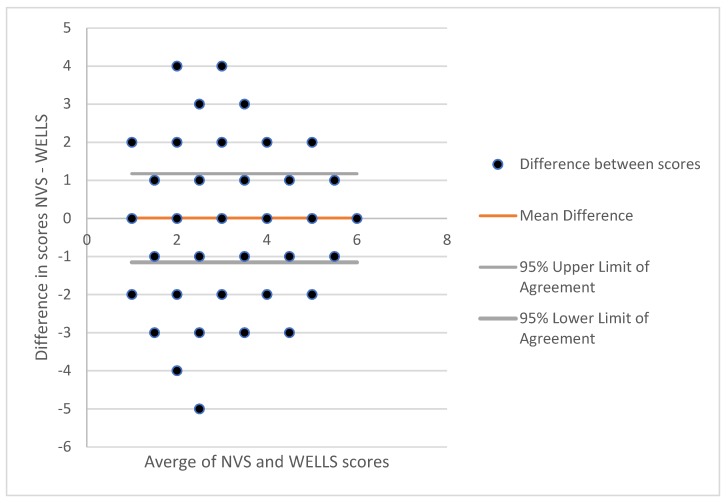
Bland-Altman plot analyzing the agreement between Newest Vital Signs and Water Environmental Literacy Level Scale (WELLS).

**Table 1 ijerph-16-00881-t001:** Characteristics of the sample and the construct validity between these characteristics and both scales.

Characteristic	Univariate Distribution	WELLS ^a^	NVS ^b^
Total Sample	*n* = 869		
	**Mean (Standard Deviation (SD))**	**Correlation**	**Correlation**
Age	37.8% (11.6) ^c^	*r* = 0.010	*r* = 0.037
	**Percentage (*n*)**	**Mean (SD)**	**Mean (SD)**
Gender			
Male	41.5% (361)	5.07 (0.99)	4.93 (1.26)
Female	57.5% (500)	4.92 (1.03)	5.04 (1.17)
Prefer not to Answer	<1% (5)	4.60 (1.67)	4.60 (1.14)
Missing	<1% (3)		
Race			
American Indian or Alaska Native	<1% (1)	-	-
Asian or Asian Indian	5.0% (43)	-	-
Black or African American	6.0% (52)	-	-
Latino	6.9% (60)	-	-
Multiracial	1.6% (14)	-	-
White	77.5% (694)	-	-
Prefer not to Answer	<1% (3)	-	-
Missing	<1% (2)	-	-
Education		-	-
College Graduate	51.8% (450)	* 5.11 (0.88)	* 5.12 (1.13)
Some College	34.4% (299)	* 4.90 (1.11)	* 4.92 (1.25)
High School Graduate	12.5% (109)	* 4.71 (1.24)	* 4.71 (1.35)
Some High School	<1% (2)	* 5.50 (0.71)	* 6.00 (0)
Prefer not to answer	<1% (6)	* 4.17 (0.98)	* 4.17 (0.75)
Missing	<1% (3)		
Income			
$0–35,999	35.0% (304)	* 4.91 (1.06)	* 4.98 (1.26)
$36,000–50,999	19.2% (167)	* 4.86 (1.17)	* 4.80 (1.30)
$51,000–75,999	19.3% (168)	* 5.05 (0.96)	* 4.96 (1.24)
$76,000 or higher	24.1% (209)	* 5.14 (0.85)	* 5.20 (0.95)
Prefer not to answer	2.3% (20)	* 4.85 (1.18)	* 4.85 (1.53)
Missing	<1% (1)		
Homeowner Status			
Own	49.3% (428)	5.04 (0.97)	5.02 (1.17)
Rent	43.8% (381)	4.91 (1.05)	4.93 (1.30)
Other Arrangement	6.8% (59)	4.97 (1.24)	5.13 (0.86)
Don’t know/Not sure	<1% (1)	5.00 (0)	6.00 (0)
Prior Well			
Yes	37.2% (323)	5.00 (0.98)	5.07 (1.13)
No	62.6% (544)	4.97 (1.04)	4.94 (1.25)
Missing	<1% (2)		
Current Well			
Yes	11.4% (99)	4.99 (1.07)	5.03 (1.20)
No	78.2% (680)	4.97 (1.00)	4.99 (1.20)
Missing	10.4% (90)		
Current Septic			
Yes	23.2% (202)	5.01 (0.97)	5.01 (1.09)
No	67.8% (589)	4.97 (1.01)	5.01 (1.23)
Missing	9.0% (78)		

^a^ Numbers are mean score and standard deviation between each demographic or household variable and the Water Environmental Literacy Level Scale (WELLS). Calculations were performed with either *t*-tests or ANOVAs. A lower case *r* denotes the use of correlations to assess the relationship between two continuous. Higher mean scores mean higher environmental health liteacy. ^b^ Numbers are mean score and standard deviation between each demographic or household variable and the Newest Vital Sign (NVS). Calculations were performed with either *t*-tests or ANOVAs. A lower case *r* denotes the use of correlations to assess the relationship between two continuous. Higher mean scores mean higher health liteacy. ^c^ Total sample size is 869. Numbers are sample size and percentage for each variables. * Indicates that there is a significant difference in scores between demographic or household characteristics.

**Table 2 ijerph-16-00881-t002:** Exact wording and frequencies for health literacy items.

Exact Item Wording	% Correct (*n* = 869)
Newest Vital Sign	% Correct
1. If you eat the entire container, how many calories will you eat?	96%
2. If you are allowed to eat 60 g of carbohydrates as a snack, how much ice cream could you have?	53%
3. Your doctor asks you to reduce the amount of saturated fat in your diet. You usually have 42 g of saturated fat each day, which includes 1 serving of ice cream. If you stop eating ice cream, how many grams of saturated fat would you be consuming each day?	87%
4. If you usually eat 2500 calories in a day, what percentage of your daily value of calories will you be eating if you eat one serving?	89%
5. For the next two questions, pretend that you are allergic to the following substances: penicillin, peanuts, latex gloves, and bee stings. Is it safe for you to eat this ice cream?	89%
6. If no, why not?	88%
Average total number correct and standard deviation	5.0 (1.2)
Water Environmental Literacy Level Scale	
a. How many ppb of arsenic is safe for cooking?	97%
b. How many ppb of arsenic in water is safe for domestic use?	41%
c. Your well water test reports that your well water is not safe for drinking. What can you do?	94%
d. Your water testing result shows arsenic at 50 ppb. How many milligrams (mg) is in one liter of your water?	77%
e. Pretend that your household water contains 15ppb of arsenic. Is it safe for you and your pets to drink?	96%
f. If no, why not?	95%
Average total number correct and standard deviation	5.0 (1.0)
Scoring. One point for each correct response. Scales range from 0–6.	

MTurk: Amazon Mechanical Turk.

**Table 3 ijerph-16-00881-t003:** Sensitivity analysis that examined the influence of item b. Where item b was recoded to be more inclusive of acceptable responses (e.g., neither a nor b), or was excluded entirely.

Statistical Measure	WELLS—Initial	WELLS—Recoded Item B	WELLS—Excluded Item B
Number of items	6	6	5
Mean (SD)	5.0 (1.03)	4.8 (0.90)	4.6 (0.84)
Range	(0–6)	(0–6)	(0–5)
Cronbach alpha	0.51	0.51	0.60
Percent in top tertile	77%	77%	92%

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
