# Peer review of "Development and Validation of an Environmental Health Literacy Assessment Screening Tool for Domestic Well Owners: The Water Environmental Literacy Level Scale (WELLS)"

_ijerph, 2019, doi:10.3390/ijerph16050881_

Round 1

Reviewer 1 Report

Overview
 This manuscript aims to develop and validate an environmental health literacy  survey instrument to evaluate whether an individual can adequately interpret private well water data and information. To do this, WELLS was developed based upon Newest Vital Sign (NVS) and contains 17 six questions on comprehension, calculations and application of information. Surveys were administered in tandem for validation. 

Overall, if the additions and revisions outlined below are made (specifically improving the discussion section), this manuscript can contribute to the literature on the development and validation of  environmental  health literacy assessment tools.

Strengths

-       The EHL among private well water owners is a critical audience to target.

-       It is novel to validate the instrument with NVS.

-       Creative methodology

-       Though not the best socio-demo representation, research had a good sample size

Weaknesses

Introduction

-       In general, the effort is oversimplifying what EHL is and what it looks like. It is understood that for this type of effort, a survey can be used as an initial screening tool to identify populations that are in need of further assistance. Authors should acknowledge that this is in fact a screening tool; and not a true detector of EHL; and clearly state, from the beginning, the additional factors that can/should be measured, understood and determined to fully understand the spectrum of an individual’s EHL.

-       Specify the number of households the water utility needs to be serving to be considered under SDWA.

-       State that currently most water utilities are not able to meet the SDWA. This is not only important to mention in the introduction; it is critical to acknowledge how EHL related to drinking water, in general (regardless of source) is limited. In discussion, you could discuss expanding this to other audiences.

-       The statistics listed on page 2, lines 56-69 should be prefaced (or it should at least be mentioned) by the limitations associated with those types of standardized tests. Since they are designed by experts and not informed those who would take the exams, they lack of tangible scenarios and do not account for free-choice or informal learning.

Methods

-       It is not clear how the survey administration platform MTurk recruits participants or what motivates individuals to be part of MTurk. More details on this recruitment and survey distribution platform are needed.

-       It is understood that the goals were to validate, but why would authors not test the instrument with a subset of actual private well owners?  This needs to be explained. Yes, 23 in-person written surveys were administered to a population with a higher %, but this was not sufficient. 

-       Units used in the excerpt used about arsenic concentrations were not defined and this is a potential limitation of the study design. Ppb or mg/L (which typically ppb is stated as micrograms per liter and ppm is milligrams per liter) was not defined and are most likely acronyms not traditionally seen by members of the general public like grams, calories or days.

Discussion

-       Please elaborate the discussion section to lessen the knowledge gap about EHL and private well owners. See comments below and address to improve discussion section.

-       Why do authors think that only 41% got the following question correct: “How many ppb of arsenic in water is safe for domestic use? This should be discussed, perhaps in the context of general public’s current numeracy literacy and/or environmental knowledge.

-       In the Introduction, it is stated, “Well water stewardship surveys report that the majority of well owners do not test their well water even though they are concerned about contaminants”. This previous finding needs to be revisited. The fact that many private well owners do not know that they even have to test their private wells is critical. Elaborate on this! Why do they not think to test? Perception? Resources? The authors should discuss how this and the fact that there was no statistically significant difference between individuals “living with a well or septic tank on their private property” and individuals who have not. The authors should also discuss how they may make modifications to the WELLS based on this evidence.

-       What knowledge gap is WELLS filling? If we already know that people are not aware about testing their private wells for contaminants in the first place, how will WELLS be used? After a treatment system is in place? After a well is initially tested to determine if the family understands the next steps? It is critical for the authors to set the stage for the application for WELLS and why one should administer this survey. What do you do if someone performs poorly on WELLS?

Author Response

Below, we provide a response to each concern raised by each reviewer. We have incorporated additional references and re-written the document based on these suggestions. We have expanded our Discussion. We thank the reviewers for their suggestions as they have greatly improved this paper.

Reviewer 1 Concerns & Responses

R1 Concern 1: Oversimplifying what EHL is and what it looks like. Authors should acknowledge that this is in fact a screening tool; and not a true detector of EHL; and clearly state, from the beginning, the additional factors that can/should be measured, understood and determined to fully understand the spectrum of an individual’s EHL. 

Response R1 Concern 1: Thank you. We agree that the EHL tool would be for screening and not a true detector. We have made edits in the Title, Abstract, Introduction and Discussion to clarify this point.

R1 Concern 2: Specify the number of households the water utility needs to be serving to be considered under SDWA.

Response R1 Concern 2: We have clarified that the SDWA applies to all public water systems that have at least 15 service connections or serve at least 25 people per day for 60 days of the year.

R1 Concern 3: State that currently most water utilities are not able to meet the SDWA. This is not only important to mention in the introduction but in the discussion, you could discuss expanding this to other audiences.

Response R1 Concern 3: Excellent suggestion. We have added this point to our discussion.

R1 Concern 4: The statistics listed on page 2, lines 56-69 should be prefaced (or it should at least be mentioned) by the limitations associated with those types of standardized tests.

Response R1 Concern 4: We have expanded in our Introduction more specific details on the results and meanings for the NAAL study cited in the above-referenced lines. We have also added details to the Limitations section of this revision.

R1 Concern 5: More details on this recruitment and survey distribution of the MTurk platform are needed.

Response R1 Concern 5: We have added substantially more details and citations to explain the MTurk participation and recruitment.

R1 Concern 6: It is understood that the goals were to validate, but why would authors not test the instrument with a subset of actual private well owners? This needs to be explained. Yes, 23 in-person written surveys were administered to a population with a higher %, but this was not sufficient.

Response R1 Concern 6: Thank you for pointing out this issue. We did have a small sample of private well owners review the WELLS scale before we conducted the survey. In addition, staff from local, regional and state domestic well program officers reviewed the scale prior to implementation. We have included this additional step in the revision.

R1 Concern 7: Units used in the excerpt used about arsenic concentrations were not defined and this is a potential limitation of the study design. Ppb or mg/L (which typically ppb is stated as micrograms per liter and ppm is milligrams per liter) was not defined and are most likely acronyms not traditionally seen by members of the general public like grams, calories or days.

Response R1 Concern 7: We opted to use the units that are listed by the Oregon Health Authority in their outreach material for this excerpt (ppb and mg/L). While we define mg/L in terms of ppb in the WELLS, it is true that the general public might not be as familiar with these units. Incorporating definitions and explanations for these units may further reduce variation between NVS and WELLs. We have included this as an additional study limitation in the revision.

R1 Concern 8: Please elaborate the discussion section to lessen the knowledge gap about EHL and private well owners. See comments below

Response R1 Concern 8: We have elaborated our discussion. See point 9 below.

R1 Concern 9:  Discuss why the authors think that only 41% got the following question correct: “How many ppb of arsenic in water is safe for domestic use? This should be discussed, perhaps in the context of general public’s current numeracy literacy and/or environmental knowledge.

Response R1 Concern 9: This is a good question. There are several possible factors that could contribute to this response rate. For example, the general public may not recognize the meaning of the word “domestic use” in this context, the unit of measure (ppb) may be new concentration value, and the general public tends to have low numeracy literacy. However, the purpose of this scale is reading comprehension and numeracy not knowledge of terminology. Therefore, we mimicked the NVS by not defining units of measures. We concurrently tested the NVS with WELLS and noticed a similar incorrect response in similar type of questions. We include this as a limitation in the revision and explain that knowledge influences comprehension.

R1 Concern 10: In the Discussion, elaborate on the fact that many private well owners do not know that they even have to test their private wells is critical. Why do they not think to test? Perception? Resources? The authors should also discuss how they may make modifications to the WELLS based on this evidence.

Response R1 Concern 10: We apologize for the confusion and clarified this sentence in the revision. Namely, well owners do not test their well even if they know that hazards are present. We have expanded reasons why with supporting citations in the Discussion.

R1 Concern 11: What knowledge gap is WELLS filling? If we already know that people are not aware about testing their private wells for contaminants in the first place, how will WELLS be used? After a treatment system is in place? After a well is initially tested to determine if the family understands the next steps? It is critical for the authors to set the stage for the application for WELLS and why one should administer this survey. What do you do if someone performs poorly on WELLS?

Response R1 Concern 11: This study is a start to develop a screener to quickly identify persons with low vs high EHL using an example of material that is relevant to the Healthy Home initiative which required individual-level health promotion. There are programs for domestic well safety in the US that have some funding to assist homeowners with treatment of wells. However, they have limited funds and want to direct these resources to individuals most at need. A screener  for EHL can identify which individuals might need more assistance and who would benefit more from the direct intervention from a well program.

Reviewer 2 Report

Table 1.  col 3 row 2.  explain the significance of the lower case r. 

line 260. sentence beginning with 'Scree'  is unclear.

figure 2.  Use WELLS throughout, introduction of EHL in the figure is confusing.

line 284.  sentence beginning with 'There' is unclear.

line 293 sentence beginning with 'We' is unclear.

line 320 replace Mturk with mTurk

Author Response

Below, we provide a response to each concern raised by each reviewer. We have incorporated additional references and re-written the document based on these suggestions. We have expanded our Discussion. We thank the reviewers for their suggestions as they have greatly improved this paper.

Reviewer 2 Concerns & Responses

R2 Concern 1: Table 1.  col 3 row 2.  Explain the significance of the lower case r.  

Response R2 Concern 2: We did not include an explanation of the lower case r in this table. Thank you catching this oversight and we have added an explanation to the table.

R2 Concern 2: line 260. Sentence beginning with 'Scree'  is unclear.

Response R2 Concern 2: We have clarified several sentences in this section pertaining to the scree plots.

R2 Concern 3: figure 2.  Use WELLS throughout, introduction of EHL in the figure is confusing.

Response R2 Concern 3:  Thank you for noticing this error. The figure has been correctly labelled.

R2 Concern 4: line 284.  Sentence beginning with 'There' is unclear.

Response R2 Concern 4: We have clarified these two sentences.

R2 Concern 5: line 293 Sentence beginning with 'We' is unclear.

Response R2 Concern 5: We have clarified this sentence.

R2 Concern 6: line 320 replace Mturk with mTurk

Response R2 Concern 6:  We have standardized the acronym to be in accordance with other papers (MTurk) and we have made sure to use consistently in our revision.

Reviewer 3 Report

This paper addresses an interesting and important question related to the interpretability of water quality information that might be presented to well owners. It is critical that well owners understand this information. However, the are two flaws that make it difficult to interpret the results of the study. 

First, the study sample is drawn from users of MTurk. These users have lower income and are better educated than the general population. It seems as if this is not likely to represent the general population well, given that education will be a key factor in determining scores on this sort of test.

Second and of greater concern, the test questions do not appear to be well constructed. In particular, the question: 

How many ppb of arsenic in water is safe for domestic use?” 

Does not have a clear answer. This reviewer, for one cannot discern the correct answer, since multiple types of domestic use are discussed with different safety standards for different uses. This seems highly ambiguous and might well be a reverse discriminator with more literate readers giving incorrect answers. Furthermore, only one other question was answered incorrectly more than 6% of the time. That means the entire score depends heavily on two questions, one of which is flawed.

Author Response

Below, we provide a response to each concern raised by each reviewer. We have incorporated additional references and re-written the document based on these suggestions. We have expanded our Discussion. We thank the reviewers for their suggestions as they have greatly improved this paper.

Reviewer3 Concerns & Responses

R3 Concern 1: First, the study sample is drawn from users of MTurk. These users have lower income and are better educated than the general population. It seems as if this is not likely to represent the general population well, given that education will be a key factor in determining scores on this sort of test.

Response R3 Concern 1:  Thank you for this point. We have expanded our discussion of MTurk in the Methods and, in the Discussion, we have compared our findings to other studies that assessed health literacy using NVS in higher-educated samples.

R3 Concern 2: The test questions do not appear to be well constructed. In particular, the question: 

“How many ppb of arsenic in water is safe for domestic use?” Does not have a clear answer. This reviewer, for one cannot discern the correct answer, since multiple types of domestic use are discussed with different safety standards for different uses. This seems highly ambiguous and might well be a reverse discriminator with more literate readers giving incorrect answers. Furthermore, only one other question was answered incorrectly more than 6% of the time. That means the entire score depends heavily on two questions, one of which is flawed.

Response R3 Concern 2:  The WELLS scale describes acceptable water uses for different arsenic concentrations. The answer to this particular question is <500 ppb, because at 500 ppb water is no longer safe for any domestic use (only 1 answer is correct). There were two items from each scale NVS and WELLS that had lower percentage of correct responses. This finding is similar to other studies that used NVS assessment in higher educated samples. We have expanded these details in the Discussion. In addition, we did pre-test these wordings in a sample of well owners. We highlight these details in our methods. However, we agree with the reviewer and we have further discussed our choice of wording and suggestions for improvement. The intention of this particular question was directed towards reading comprehension and mirrors similar questions in the NVS. That said, knowledge of terminology (e.g. domestic use, ppb) may improve reading comprehension. We discuss future improvements for the survey in the Discussion.

Round 2

Reviewer 3 Report

I find the response to concern 2 unsatisfactory for the following reasons:

1.    I would not consider water that is potentially unsafe to drink, cook with, or irrigate a garden with to be “safe for domestic use”. As an expert in this field, after reading the table on arsenic safety, I would consider the most correct answer to be 10 ppb, which is the level safe for ALL domestic use. The only unambiguous way to word a question along these lines would be to ask: How many ppb of arsenic makes water unsafe for any domestic use?

2.    The authors state, “There were two items from each scale NVS and WELLS that had lower percentage of correct responses.” I must be missing something. According to Table 2, question b was answered correctly by only 44% of respondents. The next lowest score on the WELLS questions was 77%. The lowest score on the NVS questions was 51%.

Fixing this is daunting given that, in my view the question is fundamentally flawed. I suggest that you re-analyze your data with 10 ppb as the correct answer rather than 500 ppb. You might well find that those who answered 10 ppb scored higher on the NVS questionnaire.

Author Response

Thank you reviewing our revision so promptly. We appreciate the addition opportunity to re-submit. Please see our responses to the recent critique from one reviewer.

Please let us know what else we can provide to assist with your determination for acceptance to your journal.

Response to Reviewer Comments

Reviewer Concern: Item two is still ambiguous. One way to re-word would be to say “How many ppb of arsenic makes water unsafe for any domestic use?”

Response: Thank you for this suggestion. We have added it to our Discussion.

Reviewer Concern: Original reviewer concern was that there was only 1 item in our WELLS scale that was driving the variance. We responded that there were two items in our WELLS scale that contributed to the variance. The review restated that our original response was unclear.

Response: We apologize for the misunderstanding.  There were two items in the WELLS scale that we referred to – item 2 with only 41% responding correct and item 4 with only 77% responding correctly. The additional three items had 95% or more of the sample respond correctly. We have correctly state this fact in the Discussion section.

Reviewer Suggestion: Re-analyze data with 10ppb as correct response for item 2.

Response: We did re-analyze our data although there was not a specific response option for 10ppb. We reanalyzed our data in two ways 1) leaving in question item 2 but allowing the response option – “neither a nor b” as the correct response to allow for the 10ppb answer and 2) dropping question item 2 from our scale analysis. When we re-analyzed the scale both ways, we can report similar distributional properties with the original scale analyses (see table below plus). We recompute factor analyses and all scales revealed 1 factor. We also compared our re-analyzed scales with the Newest Vital Sign and all showed similar positive, strong relationship with Newest Vital Sign. Additionally, we observed similar relationships and statistical significance with sociodemographic characteristics where people with lower health literacy where more likely to have lower household income, lower educational attainment, or self-identify as Latino. These results are now included in the revision, along with additional discussion about this particular item and suggestions for future research and/or scale improvement.

WELLS

WELLS Recoded Item 2

Wells Removed item 2

Number of items

6

6

5

Mean (SD)

5.0 (1.03)

4.8 (0.90)

4.6 (0.84)

Range

(0-6)

(0-6)

(0-5)

Cronbach alpha

0.51

0.51

0.60

Percent in top tertile

77%

77%

92%